# Observing the universal screening of a Kondo impurity

C. Piquard [1], P. Glidic [1], C. Han[2], A. Aassime[1], A. Cavanna [1], U. Gennser [1], Y. Meir [3], E. Sela[2], A. Anthore [1,4] ✉ & F. Pierre[1] ✉

The Kondo effect, deriving from a local magnetic impurity mediating electron-electron interactions, constitutes a flourishing basis for understanding a large variety of intricate many-body problems. Its experimental implementation in tunable circuits has made possible important advances through well-controlled investigations. However, these have mostly concerned transport properties, whereas thermodynamic observations - notably the fundamental measurement of the spin of the Kondo impurity - remain elusive in test-bed circuits. Here, with a novel combination of a 'charge' Kondo circuit with a charge sensor, we directly observe the state of the impurity and its progressive screening. We establish the universal renormalization flow from a single free spin to a screened singlet, the associated reduction in the magnetization, and the relationship between scaling Kondo temperature and microscopic parameters. In our device, a Kondo pseudospin is realized by two degenerate charge states of a metallic island, which we measure with a non-invasive, capacitively coupled charge sensor. Such pseudospin probe of an engineered Kondo system opens the way to the thermodynamic investigation of many exotic quantum states, including the clear observation of Majorana zero modes through their fractional entropy.

The Kondo model has proved to be an essential framework for the understanding and engineering of unconventional behaviors that develop in strongly correlated systems[1–4]. It underpins current insights into a variety of promising phenomena, from the emergence of exotic non-abelian particles[5–7] to heavy fermions[1,4] and high-$T_c$ superconductivity. The central element of this model is a local, energy degenerate 'Kondo' spin that effectively mediates interactions between itinerant electrons: As the temperature goes down, an initially weak antiferromagnetic coupling of the Kondo impurity with the spin of the electrons progressively grows, thereby giving rise to strong electron-electron correlations. The Kondo model and its variants are the continuing focus of a huge body of theoretical[8,9], numerical[10], and experimental works[11–16]. While the complexity of bulk materials impedes the data-theory comparison[17], a major step forward was made

in 1998 with the experimental implementation in nano-circuits of tunable Kondo impurities, from two degenerate spin states of a quantum dot[11,12]. However, experimental investigations of such Kondo circuits have essentially relied on their transport properties, which are inherently non-equilibrium quantities, whereas thermodynamic properties of primary interest remain elusive (see ref. 18 for a charge compressibility measurement). In particular, due to the difficulty in measuring a single elementary magnetic moment, the spin of a Kondo impurity has not been possible to probe so far, in spite of its central role.

The present work overcomes this obstacle, demonstrates the universal screening of a Kondo impurity and provides a thermodynamic window into the underlying many-body physics with a 'charge' pseudospin implementation[19]. As the role of the Kondo spin is

[1]Université Paris-Saclay, CNRS, Centre de Nanosciences et de Nanotechnologies, 91120 Palaiseau, France. [2]Raymond and Beverly Sackler School of Physics and Astronomy, Tel Aviv University, Tel Aviv 69978, Israel. [3]Department of Physics, Ben-Gurion University of the Negev, Beer-Sheva 84105, Israel. [4]Université Paris Cité, CNRS, Centre de Nanosciences et de Nanotechnologies, F-91120 Palaiseau, France. ✉e-mail: anne.anthore@c2n.upsaclay.fr; frederic.pierre@cnrs.fr

here played by two charge states, it can be sensitively measured with a capacitively coupled detector[20–23] (see refs. [24–26] for first measurements of a 'charge' Kondo impurity). Such thermodynamic charge probe permits us to investigate the primary (1-channel) Kondo effect with 'charge' Kondo circuits. In contrast, with multiple contacts required for transport characterizations, a different physics emerges in these circuits[27–29]. One noteworthy challenge to achieve a full picture is the logarithmic spread of the Kondo crossover, which extends over many orders of magnitude in $T/T_K$ with $T_K$ the scaling Kondo temperature. Here, it is addressed through the particularly broad field-effect tunability of charge Kondo circuits, allowing for large variations of $T_K$. With this approach, we observe the crossover experienced by a Kondo impurity as the temperature $T$ is lowered, from an asymptotically free (charge pseudo-) spin-1/2 to a screened singlet. The full Kondo screening is first evidenced from the saturation of the charge pseudospin susceptibility at low temperature. The complete universal crossover predicted by theory is then confronted with the temperature evolution of the Curie constant, which is considered to provide a measure of the effective spin[30]. This comparison also informs on the relation between $T_K$ and device parameters. Finally, we determine the screened Kondo (charge pseudo-) spin by measuring its fully polarized value, whose universal evolution is controlled by the ratio between Zeeman (charge states) energy splitting and Kondo temperature $T_K$.

The 'charge' Kondo mapping proposed by Matveev[31,32] is implemented in the device shown in Fig. 1a. In this mapping, the local magnetic impurity of the original Kondo model is replaced by a Kondo pseudospin of 1/2 ($S = \{\Downarrow, \Uparrow\}$) made of the two quantum charge states of lowest energy of a metallic island (bright central part), which can be tuned to degeneracy with a plunger gate voltage ($V_{pl}$). Such a reduction of the charge degree of freedom necessitates low temperatures and voltages with respect to the charging energy $E_C = e^2/2C$ ($e$ the elementary electron charge, $C$ the geometrical capacitance of the island), such that the other charge states are effectively frozen out. In practice, $E_C \simeq 39\,\mu eV \simeq k_B \times 450$ mK thus sets a high energy cutoff for Kondo physics. The important role of the magnetic field in the spin Kondo model is played by the detuning $\delta V_{pl}$ from charge degeneracy. It induces an energy difference $2E_C\delta V_{pl}/\Delta$ ($\Delta$ the plunger gate voltage period) between the pseudospin charge states, which mimics the Zeeman splitting of a Kondo magnetic impurity. Whereas the Kondo model involves a spin-exchange interaction, the island's charge (the impurity pseudospin) is not coupled to the real spin of electrons. Matveev instead introduced an electron localization pseudospin of 1/2 ($s = \{\uparrow, \downarrow\}$) that labels wave functions according to their position, either within ($\downarrow$) or outside ($\uparrow$) of the island. This localization pseudospin description requires an effectively continuous electronic density of states in the metallic island (in contrast with small quantum dots with discrete energy levels). Failing that, an electron state outside of the island ($\uparrow$) could end up without a matching state of identical energy inside it, and thus no associated pseudospin ($\downarrow$). In this representation, each time an electron enters the island, both the island's charge pseudospin $\vec{S}$ and the electron's localization pseudospin $\vec{s}$ flip. This spin-exchange process coincides with the Kondo model (with an anisotropic coupling, as the irrelevant $S_z s_z$ coupling is absent[31]). The strength of the Kondo coupling is adjusted with a tunable quantum point contact (QPC, colored red) controlling the connection to the island, and characterized by the electron transmission probability $\tau$ across the QPC. Tuning the QPC to a larger $\tau$ increases the scaling Kondo temperature, thereby opening access to lower $T/T_K$. Note however that non-universal behaviors involving the high energy cutoff $E_C$ might develop at high $\tau$, upon reaching $k_B T_K \sim E_C$ (ultimately, at precisely $\tau = 1$, the charge quantization completely vanishes[32,33] and there is no Kondo effect). Note also that the real spin of the electrons here constitutes a conserved quantum number effectively doubling the number of distinct electronic channels coupled to the metallic island. In practice, a strong magnetic field breaks the real spin

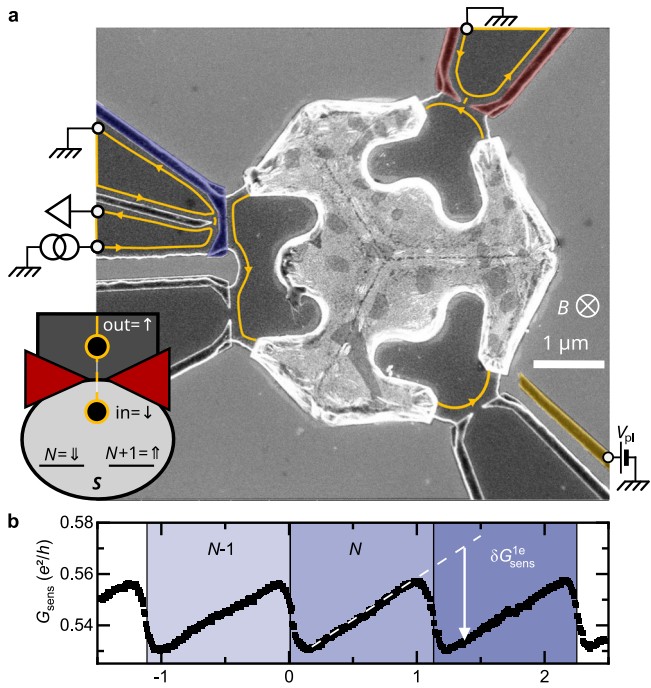

**Fig. 1 | Metal-semiconductor charge Kondo device with a charge sensor.**
**a** Colored e-beam micrograph of the measured device. A micron-scale metallic island (brighter) is connected through a tunable quantum point contact (QPC) formed by field effect using split gates (red, top-right) in a two-dimensional electron gas (2DEG, darker gray areas delimited by bright edges). The island's charge $Q$ is controlled with the plunger gate voltage $V_{pl}$ (orange, bottom-right), and measured with a capacitively coupled sensor separated by a `barring' gate (blue, left). The sensor consists of a constriction formed near the tip of the lateral gate (uncolored). The current propagates along the edges of the 2DEG set in the integer quantum Hall regime ($B \simeq 5.3$ T, filling factor $\nu = 2$), as depicted by orange lines with arrows (reflected inner channel not shown). A schematic (bottom-left) illustrates the charge Kondo mapping. **b** Sensor calibration with a weakly connected island, implementing a single-electron box. The change $\delta G_{sens}^{1e}$ in sensor conductance $G_{sens}$ for an additional $1e$ charge is given by the periodic jumps when sweeping $V_{pl}$, which are associated with discrete increments in the mean number $<N_{isl}>$ of electrons in the island. Source data are provided as a Source Data file.

degeneracy, in particular at the QPC, hence allowing to couple the spin-polarized electron channels one at a time. The primary (1-channel) Kondo effect of present interest is realized with a single electronic quantum channel (non-degenerate in real spin) connected to the island through the QPC.

The sample is nanofabricated on a Ga(Al)As heterojunction hosting a two-dimensional electron gas (2DEG) buried 90 nm below the darker surface delimited by etch-defined edges appearing bright in Fig. 1a. It is installed in a dilution refrigerator with heavily filtered and thermalized measurement lines[34], cooled down to an electronic temperature $T \simeq 9$ mK obtained from on-chip noise thermometry ("Methods" section). It is also immersed in a strong perpendicular magnetic field ($B \simeq 5.3$ T) corresponding to the integer quantum Hall effect at filling factor $\nu = 2$. In this regime, the current propagates along two quantum Hall edge channels. The outer one is schematically shown as orange lines with arrows indicating the propagation direction, while the irrelevant inner one, reflected at all QPCs, is not shown. This edge channel is in essentially perfect electrical connection with the thermally annealed AuGeNi metallic island, such that the Kondo coupling strength is entirely determined by the transmission probability $\tau$ across the single connected QPC (upper-right in Fig. 1a). Note that measuring $\tau$ requires us to connect additional channels to the island, as well as the application of a large dc voltage bias ($\sim 50\,\mu V > E_C/e$) to

minimize Coulomb effects ("Methods" section). The two other QPCs controlled by uncolored metallic split gates are only used for characterization purposes and to establish the generic (QPC independent) character of our observations ("Methods" section). Indeed, in contrast with the small quantum dot implementation of the Kondo model[35,36], multiple connected channels would compete to screen the charge Kondo pseudospin[27,28,31,32], a phenomenon with profound consequences described by another model referred to as multi-channel Kondo[1–3,37]. For the data shown in the main manuscript, the path across each of these two other QPCs is disconnected.

The island's charge sensor consists in an additional QPC whose conductance $G_{sens}$ changes by capacitive coupling with the electrical potential of the island. The sensor QPC is located along the 'barring' (separation) gate colored blue in Fig. 1a, near the tip of the uncolored gate. The 'barring' gate is negatively voltage biased to create a narrow depleted region providing a galvanic isolation (barring the way) between the sensor and the nearby edge channel emanating from the island. The conversion factor between a change in $G_{sens}$ and a change in the island's charge can be straightforwardly calibrated with the connected QPC set to a weak, tunnel contact ($\tau \simeq 0.04 \ll 1$, the island then implements a single-electron box). In this tunnel regime and at low temperatures $T \ll E_C/k_B$, the mean number of electrons in the island $\langle N_{isl} \rangle$ is quantized and periodically increases, one electron at a time, when raising $V_{pl}$. The amplitude of the corresponding periodic jumps observed in $G_{sens}(\delta V_{pl})$ (see Fig. 1b) hence provides the change $\delta G_{sens}^{1e}$ (vertical arrow) associated with the addition of one electron. Note that the charge probed by $G_{sens}$ includes the linear electrostatic contribution of $V_{pl}$ mediated by the island (a much smaller direct cross-talk contribution is separately measured and compensated for, "Methods" section), which results in the usually observed saw-tooth shape of $G_{sens}(\delta V_{pl})$ instead of a Coulomb staircase (see e.g. Ref. 26). The difference $\delta N_{isl}$ in the mean number of electrons in the island with respect to the charge degeneracy point ($\delta V_{pl} = 0$) is obtained from

$$\delta N_{isl} \equiv \langle N_{isl} \rangle (\delta V_{pl}) - \langle N_{isl} \rangle(0) = \frac{\delta G_{sens}}{\delta G_{sens}^{1e}} + \frac{\delta V_{pl}}{\Delta}, \quad (1)$$

with $\delta G_{sens} = G_{sens}(\delta V_{pl}) - G_{sens}(0)$, and $\Delta$ the period in $V_{pl}$. At $|\delta V_{pl}| < \Delta/2$, the mean value of the charge Kondo pseudospin is simply given by $\langle S_z \rangle = \delta N_{isl}$. Interestingly, despite the large geometrical difference with small quantum dots where this charge detection strategy was previously implemented[20–23], we obtain a comparable conductance sensitivity per electron of $|\delta G_{sens}^{1e}| \simeq 0.04 e^2/h$ (see "Methods" section for checks with the sensor tuned to a larger sensitivity $|\delta G_{sens}^{1e}| \simeq 0.09 e^2/h$). On the one hand, the larger geometrical capacitance of the metallic island results in a smaller voltage change $e/C$ when adding one electron compared to a quantum dot, which by itself would reduce the sensitivity of the charge sensor. On the other hand, the larger island also results in a stronger capacitive coupling to the detector, which partially compensates for the lower voltage change. Moreover, the reduced impact of tuning the sensor on the bigger Kondo circuit compared to small quantum dots allows for a better detector optimization. In principle, for strong enough measurements, the charge sensor could interfere with the probed Kondo pseudospin, by projecting it. In practice, we avoid any discernible back-action notably by driving the sensor with a rather small ac voltage bias $V_{sens}^{rms} \lesssim 3 k_B T/e$ (see "Methods" section for specific tests and discussions).

In a preliminary step, we ascertain the charge detection procedure and validate that the device is described by Matveev's model where charge Kondo physics is predicted to develop. To this aim, we directly confront in Fig. 2 the charge measurements performed over a full plunger gate voltage period $\Delta \simeq 1.14$ mV (symbols) with the quantitative analytical predictions (dashed lines) available for a connected QPC

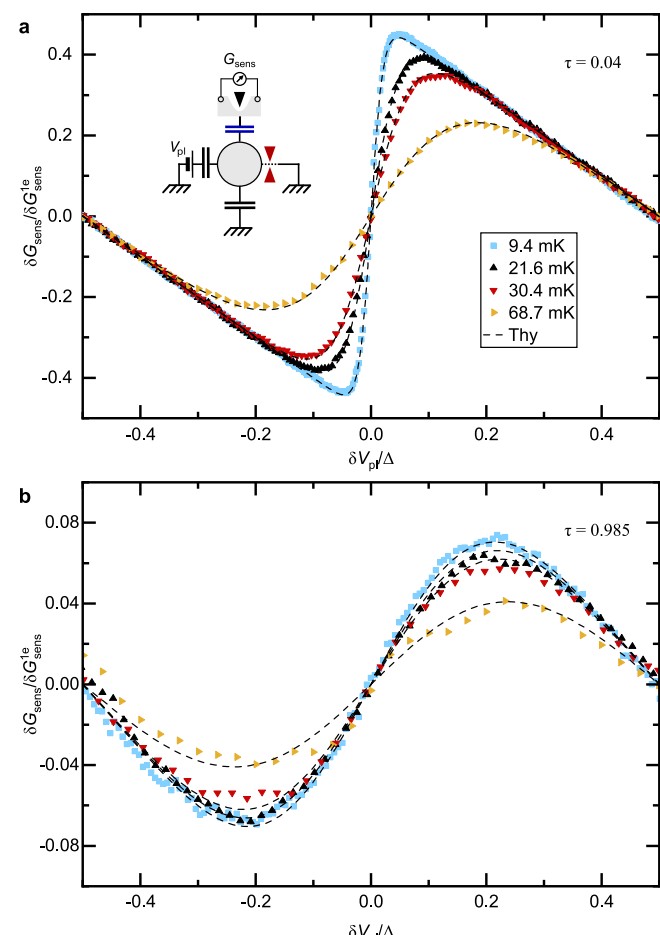

**Fig. 2 | Charge sensing in tunnel and near ballistic limits. a, b** Sensor conductance signal plotted in $1e$ step units versus plunger gate voltage difference to charge degeneracy $\delta V_{pl}$ in units of one period $\Delta$. Each set of identical symbols correspond to data points at a distinct temperature $T$ for a connected QPC set either in the tunnel ($\tau \simeq 0.04 \ll 1$, **a**) or near-ballistic ($1 - \tau \simeq 0.015 \ll 1$, **b**) limit. Measurements are plotted alongside the corresponding quantitative prediction (nearest black dashed line). Inset of **a** shows the circuit's schematic. Source data are provided as a Source Data file.

in the opposite tunnel ($\tau \simeq 0.04 \ll 1$, panel a) and near ballistic ($1 - \tau \simeq 0.015 \ll 1$, panel b) limits.

First, in the top panel (a), the displayed predictions correspond to the straightforward expression $\delta N_{isl} = \tanh(E_Z/2k_BT)/2$ for the statistical population of a pseudospin of 1/2 in the presence of the effective 'Zeeman' splitting $E_Z = 2E_C \delta V_{pl}/\Delta$, to which is added the linear plunger gate contribution $-\delta V_{pl}/\Delta$ (see Eq. (1)). Note that the charging energy $E_C \simeq k_B \times 450$ mK and the electronic temperature $T \in \{9.4, 21.6, 30.4, 68.7\}$ mK are separately characterized ("Methods" section), leaving no free parameters in the comparison. The match between data and theory shows, at our instrumental accuracy, that the sensor solely probes the island's charge, that additional charge states of the island can here be considered as frozen, and that the detector back-action is negligible even in this most sensitive tunnel configuration.

Second, in the bottom panel (b) addressing the near ballistic regime, the displayed theoretical prediction is a novel result obtained using Matveev's model of our device[32], with a second-order perturbation treatment of the back-scattering amplitude $\sqrt{1-\tau}$ valid at arbitrary $k_BT/E_C$ (see Supplementary Information for the derivation and full expression, see Eq. (4) in "Methods" section for a $O(\pi k_B T/E_C)^2$ analytical prediction very accurate except at $T \simeq 69$ mK). A remarkable,

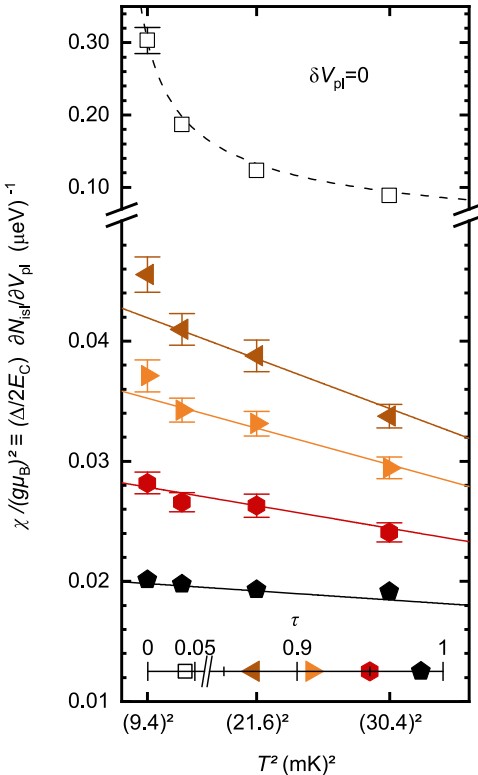

**Fig. 3 | Screened charge Kondo pseudospin at strong Kondo coupling.** The impurity charge pseudospin susceptibility measured at degeneracy ($\delta V_{pl} \simeq 0$) is plotted versus $T^2$ at $T < 40$ mK well-below the high energy cutoff ~ $E_C \simeq k_B \times 450$ mK. The continuous lines display the strong-coupling/near-ballistic quantitative predictions ($\tau$-dependent, see Methods), converging as $\propto T^2$ toward a finite zero-temperature value. A non-diverging susceptibility as $T \to 0$ signals a fully screened pseudospin. The dashed line shows the weak-coupling/tunnel quantitative prediction $\propto 1/T$ for an asymptotically free impurity. The standard error is shown when larger than symbols. Source data are provided as a Source Data file.

parameter-free agreement is observed at $\tau \simeq 0.985$ (corresponding to a back-scattering amplitude of 0.12). This agreement quantitatively validates the charge measurement in the different device regime of an almost ballistic connection to the metallic island ($1 - \tau \ll 1$), where the island's charge modulations are weak and hence relatively more sensitive to possible small artifacts such as nearby charged defects. Reciprocally, it also attests that the experimental device is adequately described by Matveev's theoretical model at high $\tau$, and in particular that the sensor back-action remains negligible. On these firm grounds, the Kondo screening of the charge pseudospin is now explored.

A central feature of the 1-channel Kondo physics is that a spin-1/2 Kondo impurity becomes fully screened at low enough temperatures $T \ll T_K$. This can be demonstrated from the low-temperature behavior of the magnetic susceptibility $\chi$ of the impurity, since a finite (non-diverging) $\chi(T \to 0)$ implies a fully screened singlet ground state[1,30]. We observe here such a screening signature of the charge Kondo pseudospin. In the present 'charge' implementation, the zero-field susceptibility $\chi/(g\mu_B)^2$ corresponds to $(\partial N_{isl}/\partial V_{pl})\Delta/2E_C$ at $\delta V_{pl} \simeq 0$. Measurements of this charge Kondo susceptibility are displayed in Fig. 3 versus $T^2$, where different symbols are associated with different Kondo couplings different $\tau$; see Supplementary Fig. 1. for $\chi$ displayed versus $\tau$. In the weak Kondo coupling limit of a tunnel QPC ($\tau \simeq 0.04$, open squares), the susceptibility increases like $1/T$ when $T$ is reduced (dashed line), as expected in the corresponding asymptotic freedom regime of an unscreened Kondo impurity ($T \gg T_K$, see Eq. (8) in "Methods" section). In contrast, for the stronger Kondo couplings implemented through a higher QPC transmission ($\tau \gtrsim 0.9$), the

susceptibility does not diverge but instead approaches a finite low temperature limit. This mere observation establishes the screening of the Kondo impurity predicted in the corresponding limit $T \ll T_K$. Quantitatively, the high transmission data closely match the parameter-free predictions derived within Matveev's model of our device (straight continuous lines, Eq. (5) in "Methods" section). The quadratic temperature scaling also corroborates the expected universal Kondo behavior at $T \ll T_K$[1]. However, the universal character does not extend to the numerical value of the $T^2$ coefficient (although it obeys the specific predictions for our device), which could be attributed[38] to $k_B T_K$ being insufficiently small compared to the high-energy cutoff $E_C$ for large $\tau$ (see Eqs. (5) and (7) and related discussion in "Methods" section).

Beyond the screened and free Kondo impurity limits, we now extend our investigation to the full, universal renormalization flow as $T/T_K$ is reduced. To this aim, we focus in Fig. 4 on the Curie constant, i.e. the susceptibility coefficient $T\chi$ (from dimensional scaling, $\chi$ without a prefactor $T$ or $T_K$ is not a universal function of $T/T_K$). For a free impurity, the Curie constant is directly related to its spin: $k_B T\chi/(g\mu_B)^2 = S(S+1)/3$ (0.25 for $S = 1/2$). More generally, the Curie constant is considered to provide a measure of the effective spin and, in particular, of the screened spin of the Kondo impurity[30]. In the left panel (a), a representative selection of 'charge' $T\chi$ data, spanning a complete range of $\tau$ (see panel b for the distribution) is displayed versus $T < 40$ mK in a linear-log scale.

First, a log-like temperature dependence characteristic of the Kondo effect is evidenced from the near-linearity of the data, except at the lowest and highest $T\chi$. However, as each tuning of $\tau$ corresponds to a different value of $T_K$, each data set seems unrelated to the others when plotted versus $T$.

Second, the central panel (b) shows a comparison of the same data, now plotted vs $T/T_K$, with the universal curve for the standard spin Kondo model (1-channel, isotropic, $S = 1/2$) obtained by combining a numerical calculation (continuous line, extracted from ref. 8) and asymptotic predictions (dashed lines, see Eqs. (9) and (10) in "Methods" section). To perform this comparison, the a priori unknown value of $T_K$ must be fixed for each $\tau$. In practice, we determine $T_K(\tau)$ such that, by construction, the data point at the lowest temperature ($T \simeq 9$ mK) lies on the predicted universal curve. The comparison in the main panel of Fig. 4b is then in the evolution as $T$ is increased for each set of identical symbols. At our experimental accuracy, we observe a good match with the universal theoretical prediction, on an explored range extending over several orders of magnitude in $T/T_K$. Although $T_K$ is a free parameter in the main panel of Fig. 4b, the underlying Kondo physics is stringently tested by further confronting this experimental $T_K(\tau)$ with predictions, see inset of b. The standard theoretical prediction for small couplings[1,31] $T_K/E_C \propto \tau^{1/4} \exp(-\pi^2/2\tau^{1/2})$ is shown as a dashed blue line, and the prediction obtained expanding upon Matveev's model for high transmissions is displayed as a dashed red line (Eq. (6) in "Methods" section). The data-theory agreement in the inset strengthens the comparison in the main panel, as the only fit parameter in the latter is seen to obey Kondo predictions in the former. Note that at large $\tau$, the Kondo temperature increases up to $k_B T_K/E_C \simeq 0.12$. Whereas non-universal deviations could develop, as pointed out in the previous paragraph, we find that they remain relatively small.

Third, for a direct, $T_K$-free comparison, the so-called $\beta$-function $\partial(T\chi)/\partial \log T$ vs $T\chi$, characterizing the universal renormalization flow of $T\chi$ is shown in Fig. 4c (continuous line) together with its analytical asymptotes (dashed lines, see expressions in "Methods" section). Such $\beta$-function are broadly used in quantum field theory to describe the running of coupling constants under a renormalization group flow. Thanks to the logarithmic derivative, the energy scale $T_K$ cancels out, and the universal character of the (monotonous) renormalization flow makes it possible to reformulate the dependence on $T/T_K$ as a dependence on the renormalized quantity ($T\chi$). In practice, data points

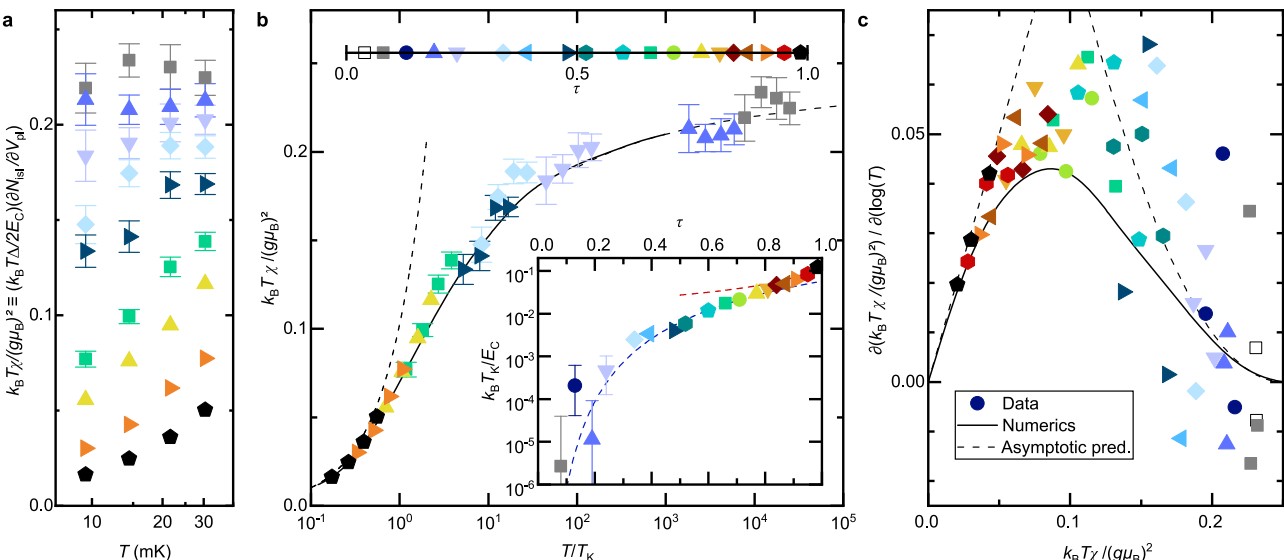

**Fig. 4 | Universal temperature renormalization flow from free to screened Kondo spin, observed on the zero-field Curie constant (spin susceptibility coefficient) $T\chi$.** Identical symbols (including in Figs. 3 and 5a) represent data points obtained for the same QPC setting (see b for correspondence symbol-$\tau$) at different temperatures. Lines are universal predictions within the standard spin Kondo model (continuous: numerical calculations, dashed: asymptotic limits). The standard error is represented when larger than symbols, except in panel **c** where it can

be inferred from the vertical spread of the data points. **a**, Measurements of $k_B T\left(\partial N_{isl}/\partial V_{pl}\right)\Delta/2E_C \equiv k_B T\chi/(g\mu_B)^2$ at $\delta V_{pl} \propto B = 0$ are plotted vs $T$ in a linear-log scale. The (near) logarithmic behavior is indicative of the Kondo effect. **b** The same $T\chi$ data points are plotted vs $T/T_K$, with $T_K(\tau)$ (see inset) obtained by matching with theory the $T \simeq 9$ mK measurements. **c** Parameter-free data-theory comparison on $\partial(T\chi)/\partial \log T$ vs $T\chi$, corresponding to the $\beta$-function characterizing the underlying Kondo renormalization flow. Source data are provided as a Source Data file.

(symbols) are obtained by a discrete differentiation of measurements at the two closest temperatures for the same device setting of $\tau$. As the experimental uncertainty on $T\chi$ ($\approx \pm 3\%$, see "Methods" section) is to be compared with its relatively small change as the temperature is incremented, the scatter of individual data points is much broader in the $\beta$-function representation (Fig. 4c) than in the Curie constant representation (Fig. 4b). Nonetheless, the similitude between the data and the numerical Kondo prediction can here be most straightforwardly appreciated.

In contrast to the above susceptibility studies performed in the absence of a Zeeman-like energy splitting of the Kondo impurity ($E_Z \ll k_B T$), we explore here the pseudospin polarization $\langle S_z \rangle$ as the degeneracy between the two charge states is lifted (Fig. 5). At large energy splitting with respect to the temperature ($E_Z \gg k_B T$), thermal fluctuations are suppressed and $\langle S_z \rangle$ corresponds to the partially screened spin of the impurity along a Kondo renormalization flow controlled by $E_Z/k_B T_K$. The universal Kondo prediction for $S_{thy}^{E_Z \gg T}(E_Z/k_B T_K)$ is shown as a continuous line in Fig. 5a[30]. Representative measurement sweeps of $\delta N_{isl} \equiv \langle S_z \rangle$, each performed for a specific setting of $\tau$ and a fixed $T$, are plotted in Fig. 5a as a function of $E_Z/k_B T_K$ (thin dashed and dotted lines indicate, respectively, $T \simeq 9$ mK and 14 mK). Here $T_K(\tau)$ is not a free parameter, but the value separately obtained from the previous data-theory comparison on the Curie constant (see inset in Fig. 4b). Only a reduced $E_Z$ interval of these measurement sweeps, highlighted with symbols, should be compared to the universal Kondo prediction. The highlighted $E_Z$ interval is limited on the low side by the requirement of a sufficiently large $E_Z/k_B T$ ratio; in practice, we set the minimum value to 5. On the high side, the comparison to the universal Kondo prediction is limited to $E_Z \ll E_C$ (equivalently, $\delta V_{pl} \ll \Delta/2$), since other charge states are otherwise relevant, which breaks the Kondo mapping. In practice, we highlight $E_Z < E_C/5$ ($|\delta V_{pl}| < 0.1\Delta$). In this interval, as shown in Fig. 5a, a data-theory difference smaller than 0.04 is observed over the wide explored parameter range, for both $T \simeq 9$ mK and 14 mK (at higher temperatures there are no data points fulfilling $5k_B T < E_Z < E_C/5$). We attribute this small discrepancy to the imperfect implementation of the theoretically

considered limit of $k_B T \ll E_Z \ll E_C$. Note that although the experimental uncertainty on $T_K$ can be important at very low $k_B T_K/E_C$ (see inset in Fig. 4b), it has little impact on the data-theory comparison, since $S_{thy}^{E_Z \gg T}$ is almost flat at the corresponding very large values of $E_Z/k_B T_K$. Hence, we experimentally establish here the predicted Kondo impurity magnetization vs $E_Z/k_B T_K$.

In a second step, we investigate the crossover from $\langle S_z \rangle = 0$ to the above universal regime as $E_Z/k_B T$ increases, for a fixed $E_Z/k_B T_K$, which is expected to exhibit specific Kondo signatures markedly different from the polarization of a free spin[8]. For this purpose, we display in Fig. 5b the Kondo impurity magnetization $\langle S_z \rangle$ normalized by the predicted universal value $S_{thy}^{E_Z \gg T}(E_Z/k_B T_K)$. Whereas for a free spin the crossover follows $\tanh(E_Z/2k_B T)$ (dashed line), as $E_Z/k_B T_K$ is reduced a logarithmic broadening is predicted to develop with the Kondo effect (continuous lines)[8]. Data points shown as identical symbols and plotted in linear-log scale versus $E_Z/k_B T$ are measurements gathered from different settings of $\tau$ (and thus different $T_K$) and different Zeeman splittings whose ratio matches the same value of $E_Z/k_B T_K \in \{0.476, 0.952, 4.76\}$. As in panel a, the collected data are limited on the high Zeeman splitting side to $E_Z < E_C/5$; however $E_Z/k_B T$ can be arbitrarily low. Similarly, we attribute the small discrepancy developing with $E_Z$ to the imperfect implementation of the theoretically considered limit $E_Z \ll E_C$. The present observation hence establishes, without any free parameter, the unconventional Kondo crossover of a partially screened impurity from a thermally averaged-out polarization to a full polarization.

By implementing a single Kondo impurity with two charge states measured by a capacitively coupled sensor, we have directly observed the central Kondo spin as it progressively hybridizes with the conduction electrons. With this approach, we demonstrated the screening of the Kondo impurity and observed the universal crossover from asymptotic freedom to the strong coupling limit on a broadly tunable and fully characterized device. Such a thermodynamic probe of a Kondo impurity opens the way to exploring the quantum back-action of a detector on a many-body state, the impurity entropy of exotic quasiparticles[23,39,40] or the entanglement negativity between Kondo

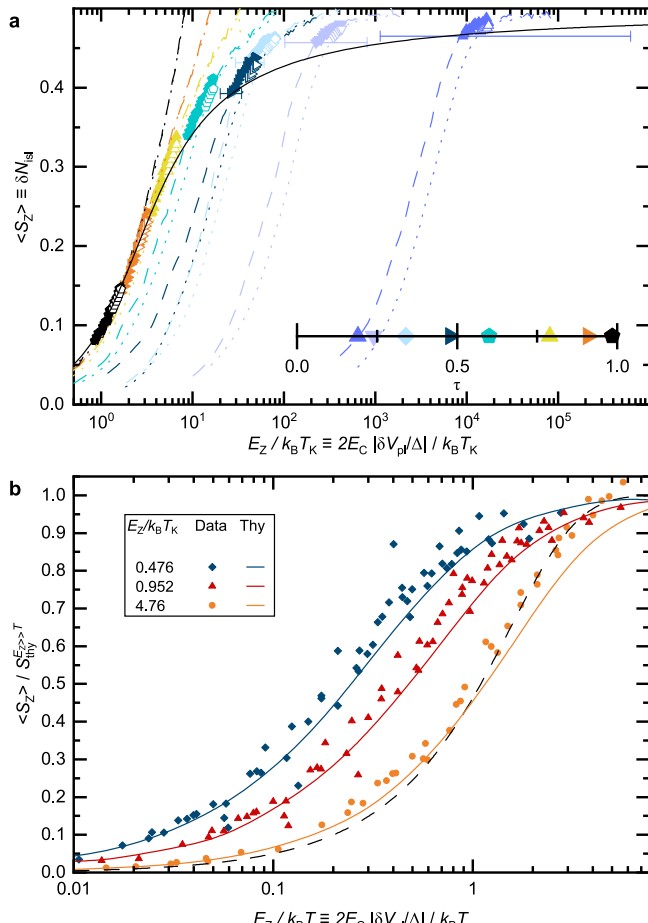

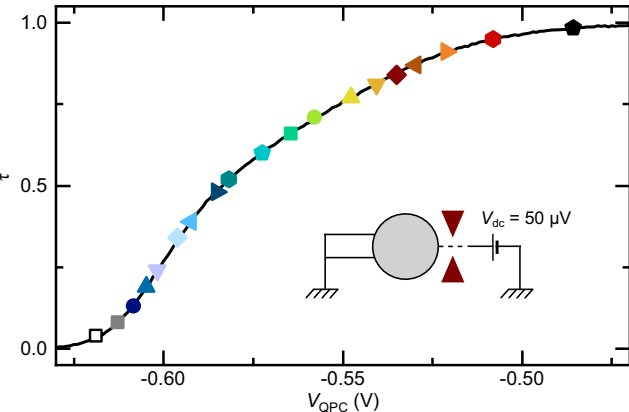

**Fig. 6 | QPC characterization.** Intrinsic transmission probability $\tau$ across the connected QPC vs applied voltage to the split gate. It is obtained from $\tau = 1/(e^2/hG - 1/2)$ with $G$ the differential conductance of the device in the configuration shown in the schematic. Symbols represent the specific settings in the main manuscript. Source data are provided as a Source Data file.

**Fig. 5 | Kondo impurity magnetization. a** $\langle S_z \rangle \equiv \delta N_{isl}$ is plotted versus the 'Zeeman' energy splitting $E_Z \equiv 2E_C|\delta V_{pl}|/\Delta$ in units of the Kondo energy scale $k_B T_K$. The Kondo prediction $S_{thy}^{E_Z \gg T}(E_Z/k_B T_K)$, in the universal limit $E_Z \gg k_B T$ where it shows the transition from free to screened spin, is represented by a thick continuous line. Measurements at $T \simeq 9$ and 14 mK are shown as dashed and dotted lines, respectively, with each color corresponding to a different $\tau$ (see inset in Fig. 4b for the separately obtained $T_K(\tau)$). Symbols (full for 9 mK, open for 14 mK) highlight data points for which a good agreement with the universal prediction is expected ($5k_B T < E_Z < E_C/5$). Horizontal error bars shown when larger than symbols represent the standard error dominated by the uncertainty on $T_K$. **b** Progressive polarization of $\langle S_z \rangle / S_{thy}^{E_Z \gg T}$ with increasing $E_Z/k_B T$ at fixed $E_Z/k_B T_K$. The thermal crossover shifts and broadens with increasing $E_Z/k_B T_K$. Gathered data points restricted to $E_Z < E_C/5$ are shown as symbols. Kondo predictions are shown as continuous lines. The free-spin prediction $\tanh(E_Z/2k_B T)$ is shown as a dashed line. Source data are provided as a Source Data file.

impurity and electron bath[41–43]. The present strategy can notably be applied to the large and continuously expanding family of Kondo-type models, hence providing a versatile platform to engineer and convincingly observe exotic states[2,3,5,6,29]. In particular, the entropy associated with the charge Kondo impurity is predicted to take a fractional value of $k_B \log 2/2$ for the two-channel Kondo model. Such a measurement would provide a clear signature for the highly debated emergence of a free Majorana mode.

## Methods
### Nanofabrication
The sample is patterned by standard e-beam lithography on a GaAlAs heterostructure forming an electron gas 90 nm below the surface, with a density of $2.6 \times 10^{11}$ cm$^{-2}$ and a mobility of $0.5 \times 10^6$ cm$^2$V$^{-1}$s$^{-1}$. The 2DEG mesa is delimited by a wet etching of ~100 nm in a H$_3$PO$_4$/H$_2$O$_2$/H$_2$O solution. The ohmic contacts are realized with a metallic

multilayer deposition of Ni(10 nm)-Au(10 nm)-Ge(90 nm)-Ni(20 nm)-Au(170 nm)-Ni(40 nm) followed by an annealing at 440°C where the metal penetrates the GaAlAs. Note that the active outer quantum Hall edge channel is found to be perfectly connected to the small metallic island at experimental accuracy (with a reflection probability below 0.1%). The gates forming the QPCs by field effect are made of aluminum deposited directly on the surface of the GaAlAs heterostructure.

### Experimental setup
The measurements are performed in a cryo-free dilution refrigerator with extensive measurement lines filtering and thermalization. Details on the fridge wiring are provided in ref. 44. Measurements of the conductance across the sample (for the device characterization) and across the charge sensor QPC are carried out with standard lock-in techniques at low frequencies, below 150 Hz, with ac excitations of rms amplitude below $k_B T/e$. Noise measurements for the thermometry described below are performed near 1 MHz with home-made cryogenic amplifiers[45].

### Electronic temperature
The electronic temperature $T$ is obtained from on-chip thermal noise measured on an ohmic contact. The conversion factor is calibrated from the linear slope of thermal noise vs temperature of the mixing chamber, at sufficiently high temperature where the difference between electron and mixing chamber temperatures is negligible. In practice, the calibration is performed above 40 mK where the high linearity of noise vs temperature attests of the good thermal anchoring of the electrons (here, differences between in-situ electronic temperature $T$ and readings of RuO$_2$ thermometers fixed to the mixing chamber develop essentially below 20 mK). The data displayed in the main manuscript are obtained at $T = \{9.4 \pm 0.3, 14.4 \pm 0.1, 21.6 \pm 0.2, 30.4 \pm 0.1, 46.2 \pm 0.1, 68.7 \pm 0.5\}$ mK, where the indicated uncertainties correspond to the temperature drifts occurring during the measurements.

### QPC characterization
The Kondo coupling strength is controlled by the bare (unrenormalized) transmission probability $\tau$ across the connected QPC shown in Fig. 6. The value of $\tau$ is obtained by setting the two other QPCs (uncolored in Fig. 1a and normally closed) well within their broad transmission plateau and by applying a dc bias of 50 μV ($>E_C$ to suppress most of the dynamical Coulomb reduction of the conductance). The device resistance is then equal to the sum of the QPC resistance

$h/\tau e^2$ and the well-defined $h/2e^2$ resistance in series. Note that changing the tuning of the two other, normally closed QPCs impacts significantly, through capacitive cross-talk, the QPC of present interest. This effect can be calibrated and is mostly corrected for, as detailed below.

## Capacitive cross-talk

The capacitive cross-talk that underpins the charge sensing mechanism also results in cross-correlations between the tunings of the different QPCs. Thanks to the distance of a few microns between constrictions, the influence of remote gates remains at a relatively small level of a few percent. Accordingly, the corresponding cross-talk can be mostly compensated for by relatively small, linear corrections calibrated separately, one pair of gates at a time. These corrections are employed for the determination of $\tau$ (see above in "Methods" section), and also to maintain a charge sensor as stable as possible with respect to changes in the tuning of the QPC, as well as during sweeps in plunger gate voltage $V_{pl}$.

## Charging energy

The experimental procedure to obtain the central charging energy $E_C \equiv e^2/2C \simeq 39\,\mu eV \simeq k_B \times 450\,mK$ combines two methods, performed with the device set to have two QPCs weakly connected to the island (Fig. 7b).

In this single-electron transistor regime, we first measure the differential conductance across the device $G_{SET}$ as a function of plunger gate voltage $\delta V_{pl}$ and dc voltage bias $V_{dc}$. The height of the resulting Coulomb diamonds displayed in Fig. 7a corresponds to $2E_C$. The red continuous lines shows the diamonds boundaries for $E_C = 39\,\mu eV$. Although very straightforward, the accuracy of this approach is mostly limited by the observed broadening of the diamond edges as $V_{dc}$ is increased.

Second, the resolution on this central quantity is refined from the width of the $G_{SET}$ peak at zero dc bias. In the asymptotic tunnel limit, where Kondo renormalization is ignored, theory predicts at $V_{dc} = 0$:

$$G_{SET}\left(\delta V_{pl}/\Delta\right) = G_{SET}(0)\,\frac{E_Z/k_B T}{\sinh(E_Z/k_B T)}, \qquad (2)$$

with $E_Z = 2E_C \delta V_{pl}/\Delta$. As illustrated in Fig. 7c for the setting $\tau \simeq 0.04$ and $T \in \{9, 22, 46\}$ mK, a very good match can be achieved between data (symbols) and Eq. (2). For each separately calibrated temperature a separate fitted value $E_C^{fit}$ is extracted, shown in the panel **d** as open and full symbols for $\tau \simeq 0.04$ and $0.08$, respectively. These fitted values provide lower bounds for $E_C$ since the residual Kondo renormalization slightly narrows the peak (by increasing the maximum conductance at degeneracy). The theoretical predictions for the effective, Kondo renormalized $E_C^*$ that is to be compared with $E_C^{fit}$ is given, in the limit of tunnel contacts, by[26,46]:

$$\frac{E_C^*}{E_C} = 1 - \frac{\sum \tau}{2\pi^2}\left(5.154 + \ln\frac{E_C}{\pi k_B T}\right) + O\left[\left(\frac{\sum \tau}{4\pi^2}\right)^2, \left(\frac{k_B T}{E_C}\right)^2\right], \qquad (3)$$

with $\sum \tau$ the sum of the transmission probabilities of all the channels connected to the metallic island (see ref. 26 for a quantitative experimental observation). Gray and black lines in Fig. 7d show the theoretical predictions of Eq. (3) for $E_C^*$ in the presence of two connected channels with the same transmission ($\sum \tau = 2\tau$) of, respectively, $\tau = 0.04$ and $0.08$, and using an unrenormalized charging energy of $E_C = 39\,\mu eV$. We estimate our experimental uncertainty on the charging energy to be of about $\pm 1\,\mu eV$.

## Charge data acquisition

The charge data are obtained from multiple repetitions of sweeps extending over several plunger gate periods. The displayed charge

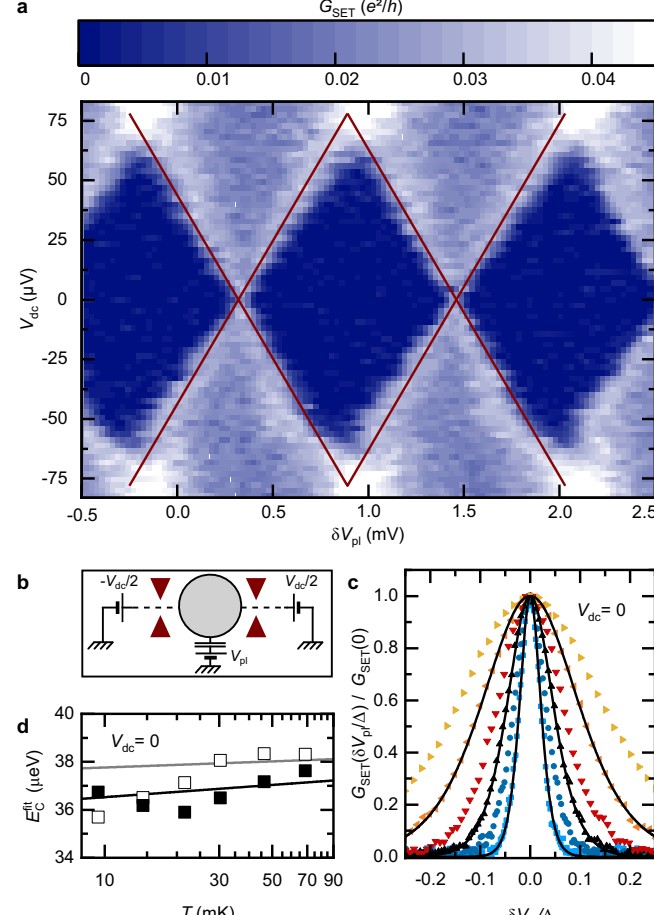

**Fig. 7 | Charging energy determination.** Charging energy extracted with the island connected by two QPCs, in the single electron transistor (SET) configuration schematically shown in **b**. Both QPCs are set to a small transmission probability. **a** The differential conductance $G_{SET}$ across the device is displayed on a color-map versus plunger gate voltage ($\delta V_{pl}$) and dc voltage ($V_{dc}$). The $E_C = 39\,\mu eV$ prediction for the Coulomb diamonds edges is shown as red straight lines. **c** The normalized conductance peak at zero dc bias (symbols, each color for a different $T$) can be fitted with the asymptotic tunnel theory (lines). The resulting $E_C^{fit}$ are shown as symbols in **d** (open and full for, respectively, lowest and slightly higher QPCs transmissions). Lines in **d** represent the corresponding predictions for a fixed $E_C = 39\,\mu eV$, also including the small Kondo renormalization effectively narrowing the peak (reducing $E_C^{fit}$). Source data are provided as a Source Data file.

data within $|\delta V_{pl}| < \Delta/2$ are hence obtained from an ensemble of at least 30 measurements. As anomalies such as nearby charge jumps can affect the integrity of the measurements, several procedures are used to automatically discard suspicious data points or sweeps (in practice, less than 10%). (1) We first check the integrity of each one period sweep. For this purpose, we integrate with $V_{pl}$ both the charge sensor conductance, as well as the absolute value of the difference between period sweep and reversed period sweep around the degeneracy point. Then we automatically discard each period sweep with a statistically anomalous integral value, defined in practice as being away from the mean by more than three times the standard deviation. (2) Second, we perform a separate statistical analysis for each individual value of $\delta V_{pl}$ on the ensemble of corresponding data points from the remaining, preserved, one period sweeps. Specific data points away from the mean by more than three times the standard deviation are automatically dropped out before the final averaging is performed.

## Experimental uncertainty

The displayed standard errors include the following independent contributions:

1. The measurement standard error, which is obtained from the statistical analysis of an ensemble of at least 30 independent data points. In practice, the standard error on sensor conductance is within $10^{-4}e^2/h$ and $2 \times 10^{-4}e^2/h$, with different specific values at different temperatures. For the susceptibility $\chi$, each of the statistically analyzed data points corresponds to an individual linear fit of the slope of the sensor conductance at the charge degeneracy point. In practice, the relative standard error on the susceptibility varies from 2% up to 8% in the most unfavorable cases.

2. The standard error on $\delta G_{\mathrm{sens}}^{1e}$ (the sensor conductance change for an addition of one electron to the island). In practice, we find a standard error within 0.6% and 0.9% of $\delta G_{\mathrm{sens}}^{1e}$ at $T \lesssim 30.4$ mK. It increases up to 2% at 68.7 mK where the range of $V_{\mathrm{pl}}$ used to extract $\delta G_{\mathrm{sens}}^{1e}$ (with a negligible thermal contribution) becomes a substantially reduced fraction of a period $\Delta$.

3. The uncertainty on temperature of typically $\pm 3$% or less (see section 'Electronic temperature'), which notably impacts the Curie constant $T\chi$ (comparably to the first, measurement standard error contribution on susceptibility).

The displayed standard errors in Figs. 3, 4 include these independent contributions. In practice, we generally find that the overall standard error is similar to the scatter between displayed data points, except notably in Fig. 5 where the vertical scatter mostly results from the challenge in being simultaneously at $E_C \gg E_Z \gg k_B T$.

## Reproducibility

The generic character of the results shown in this article is ascertained by checking their independence on (i) which specific QPC is connected to the metallic island, and (ii) the tuning of the charge sensor.

(i)  We confront in Fig. 8a, the charge measurements performed at $T \simeq 9$ mK using individually each of the three physical QPCs to connect the metallic island (distinct symbols). For a direct comparison, the three QPCs are (one at a time) tuned to the same quantitative value of $\tau$. Then we match the sensor conductance normalized by $\delta G_{\mathrm{sens}}^{1e}$. For a more thorough test, the comparison is performed over a full $\delta V_{\mathrm{pl}}$ period and for five representative values of $\tau$ spanning the full range from tunnel to ballistic (see scale bar for color code). The different symbols of the same color fall on each other, essentially within the symbols' size, showing the QPC-independent character of our results. Note that small systematic differences most likely result from the uncertainty on the experimental values of $\tau$.

(ii)  The reproducibility of the charge measurement versus charge sensor tuning is specifically checked in Fig. 8b. For this purpose, we confront the sensor conductance normalized by $\delta G_{\mathrm{sens}}^{1e}$ obtained either with the main sensor tuning used for the data shown in the main text (open squares, $\delta G_{\mathrm{sens}}^{1e} \simeq 0.0375\,e^2/h$) or with a different, more sensitive tuning (open triangles, $\delta G_{\mathrm{sens}}^{1e} \simeq 0.0937\,e^2/h$) achieved with a more negative voltage applied to the 'barring' gate (see inset). The comparison is then repeated for three different settings of $\tau$ across the QPC connected to the island, over a full $\delta V_{\mathrm{pl}}$ period. The corresponding data points are nearly indistinguishable.

## Charge sensor back-action

The charge sensor can impact the probed charge Kondo physics through the back-action induced by the charge measurement. This phenomenon is more important for stronger measurements, and therefore for larger voltages applied to determine $G_{\mathrm{sens}}$, as well as for larger tunings of the sensitivity $\delta G_{\mathrm{sens}}^{1e}$. In practice, we probe the sensor conductance using a small ac voltage of rms amplitude smaller than

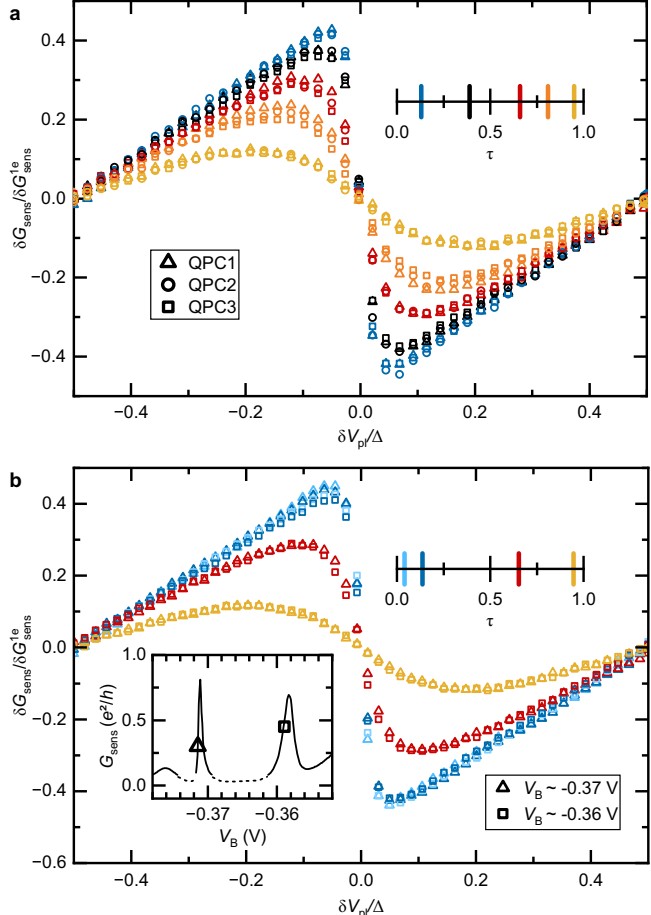

**Fig. 8 | Reproducibility.** The independence of the charge measurements on the connected QPC and on the charge sensor setting attests of the robustness and reproducibility of our results. **a** Reproducibility with different connected QPCs. Measurements at $T \simeq 9.4$ mK of the sensor conductance signal in units of the 1$e$ sensibility are plotted as symbols over one plunger gate voltage period. The different settings of $\tau$ (see color code) are each implemented on three different QPCs (different symbols) connected one at a time to the metallic island. **b** Reproducibility with different charge sensor tunings. The charge measurements shown as symbols were repeated with two sensor settings, corresponding to different voltages $V_B$ applied to the blue 'barring' gate (see inset for $G_{\mathrm{sens}}(V_B)$, the dashed line for $G_{\mathrm{sens}} \lesssim 0.1$ indicates a less reliable measurement). Squares correspond to the sensor setting used for the data in the main text, for which $\delta G_{\mathrm{sens}}^{1e} \simeq 0.0375\,e^2/h$. Triangles correspond to a more sensitive setting $\delta G_{\mathrm{sens}}^{1e} \simeq 0.0937\,e^2/h$. The normalized sensor signal at 9.4 mK is plotted versus $\delta V_{\mathrm{pl}}/\Delta$ along with three different QPC transmission $\tau$ to the metallic island (see color code). Source data are provided as a Source Data file.

$3k_B T/e$. Note however that, even at thermal equilibrium, the mere presence of a coupling to the charge sensor and/or the thermal fluctuations could be sufficient to perturb the probed system. In this section, we establish that the back-action can be neglected from (i) the negligible effect of increasing $\delta G_{\mathrm{sens}}^{1e}$, (ii) the large bias voltage margin before the 1$e$ charge step is broadened, (iii) the good fit of the charge step with the tunnel and near ballistic models ignoring the coupling to the sensor.

(i)  We find that the charge data are indistinguishable when increasing by a factor of 2.5 the sensor 1$e$ sensitivity, including in the most sensitive regime of a tunnel-coupled island (see Fig. 8b). If the back-action was not negligible and controlled by the charge detector sensitivity, a difference should have developed.

(ii)  We added a dc bias voltage to the small ac signal used to measure the sensor conductance to determine the back-action threshold.

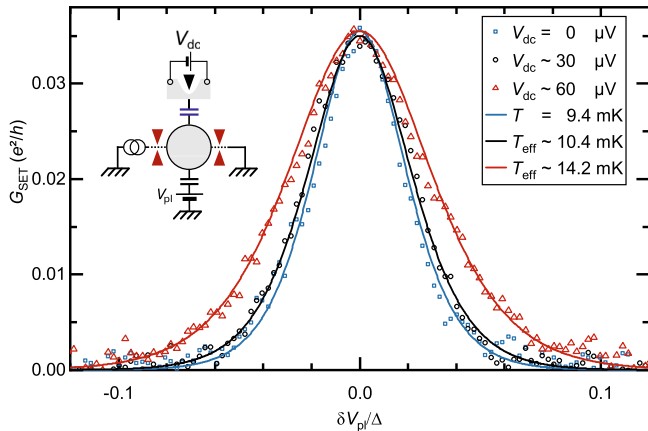

**Fig. 9 | Charge sensor back-action vs dc bias.** The conductance through the island in a SET configuration, with two connected QPCs in the tunnel regime (see schematic), is displayed as symbols versus $\delta V_{pl}/\Delta$ for different dc bias voltage $V_{dc}$ applied to the charge sensor constriction, at $T \simeq 9.4$ mK. The peak width starts to broaden for a relatively large $V_{dc} \gtrsim 30$ μV. The displayed effective temperature $T_{eff}$ are fits using the temperature as a free parameter. Source data are provided as a Source Data file.

In practice, we chose to probe not the charge (the charge sensor is impacted by the application of large biases) but the width of the peak displayed by the conductance $G_{SET}$ across the island connected by two QPCs both in the tunnel limit (see schematic in inset). In this SET regime, the peak width of $G_{SET}(\delta V_{pl})$ is proportional to the temperature if the sensor can be ignored (and in the absence of other artifacts such as nearby charge fluctuators). We find at the most sensitive, lowest temperature $T \simeq 9$ mK that a dc voltage of nearly 30 μV applied to the sensor is required to induce a visible change in $G_{SET}$ peak width (see $T_{eff}$ values for the corresponding effective temperature). This is a much larger sensor drive than both the applied ac excitation amplitude of 2 μV$_{rms}$ and the thermal fluctuations $k_B T/e \simeq 0.8$ μV.

(iii) The quantitative, parameter-less match with theory ignoring the sensor back-action further establish that it is negligible. Such comparisons are shown in Fig. 2 for the measured charge, and in the good match between blue symbols and blue line in Fig. 9 for $G_{SET}$.

## Charge Kondo circuit predictions at large QPC transmission

In this section we recapitulate the theoretical predictions at $1 - \tau \ll 1$ used to compare with the data. Their derivation is provided in the Supplementary Information.

The central new prediction, directly used in Fig. 2b, is the island charge in electron charge units $\delta Q/e \equiv \delta N_{isl}$ for arbitrary $\delta V_{pl}$, obtained for any $k_B T/E_C$ and by second-order expansion in $\sqrt{1-\tau}$ of the connected QPC. For $T \lesssim 30$ mK, this prediction given Supplementary Eq. 23 is effectively indiscernible from a second order expansion in $\pi^2 k_B T/E_C$:

$$\delta N_{isl} = \frac{\delta V_{pl}}{\Delta} + \frac{\gamma\sqrt{1-\tau}}{\pi}\left(1 - \frac{(\pi^2 k_B T)^2}{3E_C^2}\right)\sin\frac{2\pi\delta V_{pl}}{\Delta} \\ + \frac{2\gamma^2(1-\tau)}{\pi^2}\left(1 - \frac{(\pi^2 k_B T)^2}{E_C^2}\right)\sin\frac{4\pi\delta V_{pl}}{\Delta}, \tag{4}$$

where $\gamma = \exp(C_E) \simeq 1.78107$ with $C_E \simeq 0.577216$ the Euler constant. The dashed lines in Fig. 2b represent $\delta N_{isl} - \delta V_{pl}/\Delta$, with $\delta N_{isl}$ given by Supplementary Eq. 23, using the separately characterized values of $T$, $\tau = 0.985$ and $E_C = 39$ μeV.

From Eq. (4), one straightforwardly obtains for $\pi^2 k_B T/E_C, \sqrt{1-\tau} \ll 1$ the charge analog of the zero-field magnetic susceptibility:

$$\frac{\chi}{(g\mu_B)^2} \equiv \frac{\Delta}{2E_C}\frac{\partial N_{isl}}{\partial V_{pl}}\left(\delta V_{pl} = 0\right) \\ \simeq \frac{1}{2E_C}\left[1 + 2\gamma\sqrt{1-\tau} + \frac{8}{\pi}\gamma^2(1-\tau)\right. \\ \left. - \frac{(\pi^2 k_B T)^2}{E_C^2}\left(\frac{2}{3}\gamma\sqrt{1-\tau} + \frac{8}{\pi}\gamma^2(1-\tau)\right)\right]. \tag{5}$$

This expression together with the separately characterized $T$, $\tau$, and $E_C$ gives the predictions displayed as continuous lines in Fig. 3. The $T^2$ approach to a low temperature fixed value of $\chi$ corresponds to the standard Kondo prediction[1] (Eq. (7)).

The theoretical expression of the Kondo temperature can be obtained by comparing Eq. (5) with the asymptotic spin Kondo prediction in the low temperature limit $\chi(T \ll T_K) \simeq n_w/4k_B T_K$, with $n_w \simeq 0.4107$ the Wilson number (Eq. (7)). This gives in the present $\sqrt{1-\tau} \ll 1$ limit:

$$k_B T_K \simeq \frac{E_C n_w/2}{1 + 2\gamma\sqrt{1-\tau} + 8/\pi\gamma^2(1-\tau)}. \tag{6}$$

This expression for the Kondo temperature is represented as a red line in the inset of Fig. 4b. Note that it is not possible based on Eq. (5) to write $k_B T\chi$ as a function of $T/T_K$ beyond the linear term. This could be explained by the fact that, in the corresponding $\sqrt{1-\tau} \ll 1$ limit, the value of $k_B T_K$ is comparable to the high energy cutoff $E_C$ thus giving rise to non-universal contributions beyond lowest order in temperature.

## Universal Kondo predictions

In this section, we recapitulate some expressions used in the main text for the Kondo behavior of the susceptibility $\chi$, the renormalization flow of the effective spin $k_B T\chi$ and its $\beta$-function, and the renormalization flow of the Kondo impurity magnetization vs $E_Z/k_B T_K$.

The magnetic susceptibility $\chi/(g\mu_B)^2 \equiv \partial\langle S_z\rangle/\partial(g\mu_B B)$ of a spin-1/2 Kondo impurity is predicted to asymptotically approach at low temperatures (see e.g. ref. 1, Eqs. 4.58, 6.31, 6.79):

$$\frac{\chi(T \ll T_K)}{(g\mu_B)^2} = \frac{n_w}{4k_B T_K}\left(1 - \frac{\sqrt{3}}{4}\pi^3 n_w^2\left(\frac{T}{T_K}\right)^2 + O\left(\frac{T}{T_K}\right)^4\right), \tag{7}$$

with $n_w = \exp(C_E + 1/4)/\pi^{3/2} \simeq 0.4107$ the Wilson number. In the opposite limit of high temperatures with respect to $T_K$, the susceptibility reads (see e.g. ref. 1, Eq. 3.53):

$$\frac{\chi(T \gg T_K)}{(g\mu_B)^2} = \frac{1}{4k_B T}\left(1 - \frac{1}{\log(T/T_K)} - \frac{\log(\log(T/T_K))}{2\log^2(T/T_K)} + O\left(\log^{-2}(T/T_K)\right)\right). \tag{8}$$

The magnetic susceptibility coefficient (Curie constant) displayed in Fig. 4a, b is defined as $k_B T\chi/(g\mu_B)^2$. The theoretical asymptotic expression at low $T/T_K$ displayed in Fig. 4b reads:

$$\frac{k_B T\chi(T \ll T_K)}{(g\mu_B)^2} \simeq \frac{n_w}{4}\frac{T}{T_K}. \tag{9}$$

The high temperature asymptotic prediction for the magnetic susceptibility coefficient (Curie constant) displayed in Fig. 4b is calculated by solving (see e.g. ref. 1, Eqs. 4.51 and 4.52):

$$\Phi(2k_B T\chi/(g\mu_B)^2 - 1/2) = \log(T/T_K), \tag{10}$$

with the function $\Phi(x)$ given by:

$$\Phi(x) = 1/2x - \log|2x|/2 + 3.1648x + O(x^2). \tag{11}$$

Without the linear term in $x$ in the above expression of $\Phi$, solving Eq. (10) leads to the expression of $\chi$ given in Eq. (8). Including the linear term increases the range of validity in $T/T_K$ of the asymptotic solution, allowing us to make contact with the numerical calculation available up to $T/T_K \approx 10^3$.

The theoretical $\beta$-function characterizing the renormalization flow of the magnetic susceptibility coefficient (Curie constant) is defined as $\beta \equiv \partial(k_B T\chi/(g\mu_B)^2)/\partial \log T$ as a function of $k_B T\chi/(g\mu_B)^2$. From Eq. (9), which is linear in $T$, the low $T\chi$ asymptote displayed as a dashed straight line in Fig. 4c is simply:

$$\beta(x \ll 1) \simeq x, \tag{12}$$

with $x \equiv k_B T\chi/(g\mu_B)^2$. From Eq. (8), keeping only the first two terms in the parenthesis, the asymptote near $k_B T\chi/(g\mu_B)^2 \approx 1/4$ displayed as a dashed line in Fig. 4c reads:

$$\beta(1/4 - x \ll 1) \simeq (1 - 4x)^2/4. \tag{13}$$

The full prediction shown as a continuous line in Fig. 4c is obtained by discrete differentiation of the numerical calculation (continuous line in Fig. 4b), averaged by Fourier filtering and completed by the asymptotic analytical predictions.

The magnetization of a Kondo impurity in the limit of low temperatures with respect to the Zeeman splitting ($k_B T \ll E_Z$, both small compared to the high energy cutoff) is predicted to follow a universal renormalization flow controlled by the parameter $E_Z/k_B T_K$, which is shown as a continuous line in Fig. 5a. This prediction can be expressed analytically, with a different equation depending on whether $x \equiv \frac{E_Z}{k_B T_K} \frac{\exp(C_E + 3/4)}{2\pi\sqrt{2}}$ is larger or smaller than 1 (see Eqs. 4.29c in Ref. 30, and Ref. 1 or the comparison with Eq. (7) at $x \ll 1$ for the numerical factor between $x$ and $E_Z/k_B T_K$):

$$S_{thy}^{E_Z \gg T}(x > 1) = \frac{1}{2} - \frac{1}{2\pi^{3/2}} \int_0^\infty dt \frac{\sin(\pi t)}{t} e^{-t\ln t + t} x^{-2t} \Gamma\left(t + \frac{1}{2}\right),$$
$$S_{thy}^{E_Z \gg T}(x < 1) = \frac{1}{2\sqrt{\pi}} \sum_{k=0}^\infty (-1)^k \frac{(k + \frac{1}{2})^{k-1/2}}{k!} e^{-k-1/2} x^{2k+1}. \tag{14}$$

## Data availability
The data used to produce the plots within this paper are available via Zenodo at https://doi.org/10.5281/zenodo.10033055. Source data are provided with this paper.

## Code availability
The codes used to produce the plots within this paper are available via Zenodo at https://doi.org/10.5281/zenodo.10033055.

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

## Acknowledgements
This work was supported by the European Research Council (ERC-2020-SyG-951451), the French National Research Agency (ANR-18-CE47-0014-01) and the French RENATECH network. We thank K. Ensslin, J. Folk and A. Mitchell for discussions.

## Author contributions
C.P. performed the experiment with inputs from A.Aa., A.An., F.P., and P.G.; C.P., A.An., and F.P. analyzed the data; C.P., A.Aa and F.P. fabricated the sample; A.C. and U.G. grew the 2DEG; C.H., E.S., and Y.M. developed the theory; C.P. and F.P. wrote the paper with inputs from all authors; A.An. and F.P. led the project.

## Competing interests
The authors declare no competing interests.
