## [Peer Review File · Nature Communications]

REVIEWER COMMENTS

Reviewer #1 (Remarks to the Author):

The authors use a QPC charge sensor to capacitively couple and investigate the thermodynamics of a charge Kondo system. This research team has made significant achievements in studying charge Kondo systems through transport measurements in the past. In this work, they made some small modifications to their devices and focused on the thermodynamics rather than electrical transport. The research method and results they present are quite interesting and can open a new avenue for thermodynamic and entropy measurements in nanoscale systems. However, I think the authors should further consider and clarify the following point before I recommend the work for publication in Nature Communications.

I acknowledge the importance of the thermodynamic measurements. However, it's not clear to me what new physics about the charge Kondo effect was provided in this work. The authors' demonstration of the charge Kondo effect in their system is not surprising given their past achievements. Although the results presented in this work are beautiful and well-consistent with the Kondo model, they do not provide new understanding of charge Kondo physics beyond prior studies. Since the charge Kondo effect has been observed and well-studied before, it's essential for the authors to address what new scientific insights into the Kondo effect their QPC charge sensors provide.

Following the point above, the title "Observation of a Kondo impurity state and universal screening using a charge pseudospin" may be a bit misleading and not clearly show the contribution of this work.

It would be helpful for the authors to clarify the role of the real spin of the electron and magnetic fields in their work. From Matveev's model, the electron spin can be viewed as a "color" for the electrons, and the system should be mapped onto a two-channel Kondo model. Additionally, since the system is set in the quantum Hall regime ($\nu = 2$) with $B = 5.3$ T, it may be important to consider how the Zeeman splitting from the real spin splitting affects their system.

The results of the universal temperature renormalization flow look interesting, but a more intuitive picture is missing for a journal with broad readerships like this. For example, the physical significance of the β -function characterization shown in Fig. 4c is not immediately apparent. The authors should provide additional explanation and context to help readers understand the importance of this figure. The authors should also further comment on the large deviation between the theory and

experiment since the current argument about the discrete temperature differentiation is not convincing.

There is a similar case for Fig. 5 when a large Zeeman-like energy splitting was applied. The evolution of the pseudo-magnetization appears to be expected, which leaves the significance of the results not immediately apparent.

Reviewer #2 (Remarks to the Author):

Re: Manuscript#: NCOMMS-23-03942-T

Observation of a Kondo impurity state and universal screening using a charge pseudospin

by C. Piquard et al.

The authors reported careful measurements of the (pseudo)spin of a Kondo impurity by using a quantum point contact (QPC, a charge detector) capacitively coupled to a charge Kondo metallic island. They found that the QPC senses well addition of a single electron charge to the island, obtaining the conversion factor from the QPC signal to the charge change. Then applying it to different (low-temperature Kondo, higher temperature weak coupling, and the crossover) regimes of the island, they observed the zero-field susceptibility (and the Curie constant) and the impurity pseudospin magnetization. The dependence of the observables on the temperature or the pseudo-magnetic field is in agreement with the theories. The Extended Data Figures support the main conclusions and are useful for future studies.

In my opinion, this work demonstrated an important step (the charge sensing for such a large metallic island) that has not been achieved previously and opens a road toward unexplored areas of quantum impurity problems and quantum transports of correlated electrons. The manuscript is well written for broad audience. Therefore I recommend the paper for publication in Nature Communications, after the authors consider a few minor comments below.

(1) As the authors point out, their charge sensor shows a sensitivity comparable to previous works on a QPC coupled to a much smaller quantum dot. It would be useful to discuss the reasoning in the main text, with counting, e.g., geometric factors or excess-charge distribution over the island.

(2) It will be useful to clarify measurement uncertainty in the data plots. Is the measurement uncertainty smaller than the data-point size in Figs. 1-5?

(3) It is understandable that the experimental data of the beta function in Fig. 4c has (relatively) large uncertainty because of the discrete temperature differentiation. Is it possible to quantify measurement uncertainty of this data set?

(4) In Figs. 5a and 5b, the measured data match well with the theory prediction. But the agreement seems less than the excellent match shown in Fig. 4b. Does this relate with the determination of Kondo temperature (implying possibility of a better determination) or some other reasoning as in Fig. 4c?

(5) The theory in Supplementary Information can be written more reader-friendly.

=====

Reviewer #3 (Remarks to the Author):

This is a very nice and detailed paper, which investigates the properties of the charge Kondo with a dedicated charge sensor. This allows the authors to investigate the effective “spin” properties (e.g. susceptibility) via the charge measurements. This is not the first attempt of this sort – the authors should cite <http://aip.scitation.org/doi/10.1063/1.123450> which also explored the charge Kondo with a charge sensor and covered the range from the closed to the open contact regime. However, the present paper is much more detailed and presents cleaner data over a broader range of parameters, which allows them to explore the scaling relations. Unfortunately, the temperature could be only changed from 10 to 30 mK before leaving the universal regime (Figure 4a), which results in each group of points being all bunched together in the key Figure 4b. This is a natural limitation of this measurement, and fortunately the scaling can still be traced via the dependence on T_{Kondo} . However, the resulting spread in Figure 4c, which involves the temperature derivative is quite large, and the agreement is only qualitative. Finally, the scaling at finite “Zeeman field” (gate voltage detuning) is very convincing.

I recommend publishing the paper with minor revisions.

Minor comments:

It would be interesting to produce a figure similar to Figure 4 of the reference above (susceptibility at zero Zeeman field vs contact transparency), perhaps in the supplementary.

The authors could consider rewriting the following text in a more readable fashion: "This validates the non-invasive island charge sensing in the present regime of weak charge modulations, more sensitive to nearby charged defects and small artifacts. This agreement also attests of the adequation between device and theoretical model at high τ ."

The meaning term "the barring gate" is not clear.

We would like to express our appreciation for the reviewers' positive appreciations and for their suggestions of improvements.

Reviewer #1

The authors use a QPC charge sensor to capacitively couple and investigate the thermodynamics of a charge Kondo system. This research team has made significant achievements in studying charge Kondo systems through transport measurements in the past. In this work, they made some small modifications to their devices and focused on the thermodynamics rather than electrical transport. The research method and results they present are quite interesting and can open a new avenue for thermodynamic and entropy measurements in nanoscale systems. However, I think the authors should further consider and clarify the following point before I recommend the work for publication in Nature Communications.

I acknowledge the importance of the thermodynamic measurements. However, it's not clear to me what new physics about the charge Kondo effect was provided in this work. The authors' demonstration of the charge Kondo effect in their system is not surprising given their past achievements. Although the results presented in this work are beautiful and well-consistent with the Kondo model, they do not provide new understanding of charge Kondo physics beyond prior studies. Since the charge Kondo effect has been observed and well-studied before, it's essential for the authors to address what new scientific insights into the Kondo effect their QPC charge sensors provide.

Following the point above, the title "Observation of a Kondo impurity state and universal screening using a charge pseudospin" may be a bit misleading and not clearly show the contribution of this work.

[Reply]

The reviewer points that the present work does not bring new information on the well-understood Kondo physics, nor on the specific charge Kondo implementation. However, it provides a direct experimental observation of what is arguably the most central feature in Kondo physics, namely the universal screening flow of a Kondo impurity.

Here we perform the first experimental observation of the full universal renormalization flow of a Kondo impurity from a free spin to a screened singlet in a tunable circuit, and the first observation of the (pseudo)spin state of a single Kondo impurity. Despite being at the heart of the Kondo effect, the predicted universal screening of a single Kondo impurity could not be established before. In addition, compared to previous efforts in bulk systems with multiple Kondo impurities, we here achieve an unprecedented precision, an unprecedented broad range of rescaled temperature T/T_K , and a measurement in a tunable circuit free from the complexity of bulk materials (such as crystal field impact on impurity spin, impurity-impurity interactions, precision and reliability of magnetic moment measurements in the dilute impurity limit, challenge in exploring a crossover extending over 5 orders of magnitude in T/T_K ...). This was made possible by implementing the Kondo impurity with circuit charge states, which we could directly probe using a capacitively coupled sensor. Looking ahead, we strongly believe that the presently implemented combination of a 'charge' Kondo circuit with a charge sensor will open the path to scrutinize exotic states, such as those predicted to emerge in Kondo-type models (Majorana, parafermions).

In response to the reviewer we simplified and refocused the title from (previous version) "Observation of a Kondo impurity state and universal screening using a charge pseudospin" to (present version) "Observing the universal screening of a Kondo impurity".

[Reviewer #1]

It would be helpful for the authors to clarify the role of the real spin of the electron and magnetic fields in their work. From Matveev's model, the electron spin can be viewed as a "color" for the electrons, and the system should be mapped onto a two-channel Kondo model. Additionally, since the system is set in the quantum Hall regime ($\nu = 2$) with $B = 5.3$ T, it may be important to consider how the Zeeman splitting from the real spin splitting affects their system.

[Reply]

We thank the reviewer for this suggestion of clarification.

Indeed, the Zeeman splitting of the real electron spin plays a key role as, in the absence of a splitting, the real spin degeneracy would effectively double the number of connected channels and a different (two-channel) Kondo physics would develop. Here, at $\nu=2$, the outer edge channel of transmission τ is spin polarized, with a corresponding conductance of e^2/h at $\tau=1$. In response to the reviewer this point is now further clarified in the manuscript (see added text page 2, right column in marked manuscript).

[Reviewer #1]

The results of the universal temperature renormalization flow look interesting, but a more intuitive picture is missing for a journal with broad readerships like this. For example, the physical significance of the β -function characterization shown in Fig. 4c is not immediately apparent. The authors should provide additional explanation and context to help readers understand the importance of this figure.

[Reply]

We thank the reviewer for suggesting this pedagogical improvement. In response to the reviewer, we have added a discussion on beta-functions. See page 5, top of right column in marked manuscript.

[Reviewer #1]

The authors should also further comment on the large deviation between the theory and experiment since the current argument about the discrete temperature differentiation is not convincing.

[Reply]

In the considered Fig. 4c, the scatter of experimental data points is indeed relatively important. This increased noise results from the relatively small difference in $T\chi$ with respect to experimental noise between two successive temperatures. As each data point in panel c corresponds to a discrete differentiation with two successive temperature (at a given τ , shown by identical symbols), the experimental noise has a relatively strong impact. Whereas a better-looking agreement would have been obtained if we had chosen larger temperature steps for the differentiation, these larger steps would have become significant with respect to the predicted non-linearity of the beta-function. The chosen procedure most directly shows the experimental uncertainty from the spread of the cloud of data points.

In response to the reviewer, we have improved our discussion on the noise level in Fig. 4c and now also display the standard measurement error on χ and $T\chi$ when larger than the symbols (see page 5, top of right column in marked manuscript and the new section in Methods regarding experimental uncertainties).

[Reviewer #1]

There is a similar case for Fig. 5 when a large Zeeman-like energy splitting was applied. The evolution of the pseudo-magnetization appears to be expected, which leaves the significance of the results not immediately apparent.

[Reply]

As we understand it, this remark of the reviewer probably echoes the first remark on looking for new understanding on the Kondo effect or on the charge Kondo implementation, for which we refer to our answer on the first point. Note also that the quantitative data-theory comparison is here free of any fit parameters (without involving the challenging beta-function) since T_K is separately extracted from the Curie constant at zero Zeeman splitting.

Another possibility is that the reviewer was asking about the physical significance of the plotted quantities. In Fig. 5a, the mean value of the (charge pseudo-)spin at a large energy splitting E_Z with respect to temperature allows us to suppress thermal fluctuations, and hence it directly corresponds to the effective moment of the partially screened Kondo impurity (controlled within the universal renormalization flow by E_Z/kT_K instead of T/T_K). In response to the reviewer, we have modified our description of the observed quantities, and we reformulated the recapitulative sentences of panels a and b.

Reviewer #2

The authors reported careful measurements of the (pseudo)spin of a Kondo impurity by using a quantum point contact (QPC, a charge detector) capacitively coupled to a charge Kondo metallic island. They found that the QPC senses well addition of a single electron charge to the island, obtaining the conversion factor from the QPC signal to the charge change. Then applying it to different (low-temperature Kondo, higher temperature weak coupling, and the crossover) regimes of the island, they observed the zero-field susceptibility (and the Curie constant) and the impurity pseudospin magnetization. The dependence of the observables on the temperature or the pseudo-magnetic field is in agreement with the theories. The Extended Data Figures support the main conclusions and are useful for future studies.

In my opinion, this work demonstrated an important step (the charge sensing for such a large metallic island) that has not been achieved previously and opens a road toward unexplored areas of quantum impurity problems and quantum transports of correlated electrons. The manuscript is well written for broad audience. Therefore I recommend the paper for publication in Nature Communications, after the authors consider a few minor comments below.

(1) As the authors point out, their charge sensor shows a sensitivity comparable to previous works on a QPC coupled to a much smaller quantum dot. It would be useful to discuss the reasoning in the main text, with counting, e.g., geometric factors or excess-charge distribution over the island.

[Reply]

We thank the reviewer for suggesting this improvement. Following the reviewer's suggestion, we have added a discussion of this point in the main manuscript, see page 3, top of right column of marked manuscript.

[Reviewer #2]

(2) It will be useful to clarify measurement uncertainty in the data plots. Is the measurement uncertainty smaller than the data-point size in Figs. 1-5?

(3) It is understandable that the experimental data of the beta function in Fig. 4c has (relatively) large uncertainty because of the discrete temperature differentiation. Is it possible to quantify measurement uncertainty of this data set?

[Reply]

The experimental error is typically comparable or smaller than the size of the data points except, most notably, in Fig. 4c (where the symbols are much smaller than the uncertainty). This could be inferred from the scatter of the data.

In response to the reviewer, we refined our data analysis to most straightforwardly extract the experimental standard error (see new dedicated section in Methods), now often displayed as error bars when larger than the symbols. Note that we find that adding error bars on the charge sensor (Fig. 2b), beta-function (Fig. 4c) and magnetization (Fig. 5b) data would deteriorate the readability of the figures whereas the experimental uncertainty could be inferred from the scatter of the points (and from the values of the different standard error contributions now detailed in Methods).

[Reviewer #2]

(4) In Figs. 5a and 5b, the measured data match well with the theory prediction. But the agreement seems less than the excellent match shown in Fig. 4b. Does this relate with the determination of Kondo temperature (implying possibility of a better determination) or some other reasoning as in Fig. 4c?

[Reply]

As noticed by the reviewer, the data/theory comparisons in Fig. 5 are without any fit parameter, since we use the Kondo temperature previously determined in Fig 4b (by matching the lowest temperature points in the main panel with theory). The uncertainty on T_K thus constitutes a source of error here (as an overall shift for each set of data points in panel a, for the E_Z/K_{BT_K} selection in panel b). However the largest uncertainties on T_K are in the regime of a weak Kondo effect where it has the smallest effect on the data-theory comparison (because of the small dependence on $T/T_K \gg 1$).

The stringent specific challenge in the comparisons shown in Fig. 5 is that when increasing the effective Zeeman splitting, we reduce the headroom until other charge states come into play, which breaks the mapping to the standard Kondo theory confronted to the data.

We attribute the relatively small but visible data-theory discrepancy to, for the most part, the imperfect implementation of the theoretically considered limit of $E_Z \ll E_C$ (and $kT \ll E_Z$ for panel a).

In response to the reviewer, this is now specifically indicated in the manuscript (see page 6, left and right columns in marked manuscript).

[Reviewer #2]

(5) The theory in Supplementary Information can be written more reader-friendly.

[Reply]

In response to the reviewer, we much expanded the text describing the theoretical steps in the (main) first section of the supplementary information.

Reviewer #3

This is a very nice and detailed paper, which investigates the properties of the charge Kondo with a dedicated charge sensor. This allows the authors to investigate the effective “spin” properties (e.g. susceptibility) via the charge measurements. This is not the first attempt of this sort – the authors should cite <http://aip.scitation.org/doi/10.1063/1.123450> which also explored the charge Kondo with a charge sensor and covered the range from the closed to the open contact regime. However, the present paper is much more detailed and presents cleaner data over a broader range of parameters, which allows them to explore the scaling relations.

[Reply]

Following the reviewer’s suggestion, we added a reference to this first attempt [27], together with the similar work by a different team [26] and another work on tunnel contacted single electron box by Lehnert et al ([28] (previous reference [35], limited to first signatures at $T_K \ll T$), see page 1, top of right column.

Note that, due to degeneracy in real spin of the connected channel in the experiment [26,27] (performed at $B=0$), the charge Kondo mapping near the charge degeneracy points would be on the alternative two-channel Kondo model (and the experiment of [17] in the many channels limit, although this does not impact the behavior in the probed regime of $T \gg T_K$).

Note also that, besides the limitations of device and temperature characterizations hinted by the reviewer, in the previous works [26,27] the charge Kondo mapping itself (at charge degeneracy) is likely to be limited by the discreteness of electronic levels in the probed sub-micron AlGaAs island.

[Reviewer #3]

Unfortunately, the temperature could be only changed from 10 to 30 mK before leaving the universal regime (Figure 4a), which results in each group of points being all bunched together in the key Figure 4b. This is a natural limitation of this measurement, and fortunately the scaling can still be traced via the dependence on T Kondo. However, the resulting spread in Figure 4c, which involves the temperature derivative is quite large, and the agreement is only qualitative. Finally, the scaling at finite “Zeeman field” (gate voltage detuning) is very convincing.

I recommend publishing the paper with minor revisions.

Minor comments:

It would be interesting to produce a figure similar to Figure 4 of the reference above (susceptibility at zero Zeeman field vs contact transparency), perhaps in the supplementary.

[Reply]

The Figure 4 in [27] focuses on a regime of a high Zeeman field where the Kondo effect is strongly suppressed (deep in the Coulomb valley, as far as possible from the charge degeneracy points). In fact, the charge Kondo mapping does not apply in that case as there are two excited charge states of identical energy (constituting a 3 states system including the fundamental charge state, instead of a two states Zeeman split spin).

Following the spirit of the reviewer’s suggestion, we plotted in the supplementary information (new Fig. S1) the charge susceptibility measured at the degeneracy point (where the Kondo effect fully develops) versus τ , and compared to the theoretical predictions obtained for this specific system in the asymptotic limits of high and low τ . This complements the figure 3 where the focus is on the temperature dependence of the charge susceptibility. This alternative SI figure showing the

susceptibility measurements vs τ is pointed out in the manuscript (see page 4, right column of marked manuscript).

[Reviewer #3]

The authors could consider rewriting the following text in a more readable fashion: “This validates the non-invasive island charge sensing in the present regime of weak charge modulations, more sensitive to nearby charged defects and small artifacts. This agreement also attests of the adequation between device and theoretical model at high τ .”

[Reply]

We thank the reviewer for pointing this out.

In response to the reviewer, these sentences have been rewritten into:

“This agreement quantitatively validates the charge measurement in the different device regime of an almost ballistic connection to the metallic island ($\tau \ll 1$), where the island’s charge modulations are weak and hence relatively more sensitive to possible small artifacts such as nearby charged defects. Reciprocally, it also attests that the experimental device is adequately described by Matveev’s theoretical model at high τ , and in particular that the sensor back-action remains negligible.”

[Reviewer #3]

The meaning term “the barring gate” is not clear.

[Reply]

We chose this calling name for the gate colored blue to emphasize that it is used to create a barrier, a galvanic isolation, barring the way between the charge Kondo island and the charge sensor.

In response to the reviewer we now clarify that barring also means here separation “the ‘barring’ (separation) gate”, that it provides “a galvanic isolation (barring the way) between ...” and barring systematically appears between inverted commas (‘barring’) to stress that this is a calling name (see page 3 left column in marked manuscript for the above quotes).

Detailed list of changes:

See marked manuscript. The new text is in red, except new headlines and new or updated references. The text removed is barred.

The data analysis has been refined to allow for the extraction of the standard error, which could result in small changes in the position of data points without any consequences on the discussions and conclusions.

References: We added the references [18, 26, 27]

Supplementary Information:

The text in section I describing the theoretical derivation of Eqs 4,5 was much extended in response to Reviewer #3.

A new supplementary section III and supplementary figure S1 were added in response to Reviewer #3.

REVIEWERS' COMMENTS

Reviewer #1 (Remarks to the Author):

In the revised manuscript, the authors have adequately addressed most of my questions, except for what new information can be obtained from their capacitively coupled charge sensor measurements. In their response, the authors argued that they "perform the first experimental observation of the full universal renormalization flow of a Kondo impurity from a free spin to a screened singlet in a tunable circuit". However, a universal Kondo screening and renormalization has in fact been demonstrated in several previous studies, including Ref. 20-21, which were conducted by the same research group, even though some prior results may focus on other aspects and/or not be as complete as this one.

As for now, I still do not see any new physical insights into Kondo impurity that have yet been or cannot be obtained using transport measurement in a non-equilibrium regime. However, I would also like to stress that I appreciate the elegance and significance of using a charge sensor to study the charge Kondo effect, and believe that this work may open a new avenue for thermodynamic and entropy measurements in nanoscale systems. The quality of both the study and the manuscript is good. Hence, I am still happy to recommend it for publication in Nature Communications.